# Management of ocular adverse reactions to antiglaucoma medications–Survey of optometrists in Ghana

Mohammed Abdul-Kabir[1]*, Prince Mintah[1], Michael Kwesi Asante[1], James Forson[2‡], Agnes Oppong[3‡], James Anin Odame[1‡], Gabriel Kwaku Agbeshie[1], Stephanie Obeng-Inkoom[4‡]

**1** Department of Optometry and Visual Science, Kwame Nkrumah University of Science and Technology, Kumasi, Ghana, **2** Dr. Bawuah Memorial Hospital, Kete-Krachi, Ghana, **3** Kumasi South Hospital, Kumasi, Ghana, **4** Optometry Unit, Korle-Bu Teaching Hospital Eye Center, Accra, Ghana

☯ These authors contributed equally to this work.
‡ These authors also contributed equally to this work.
* mabdul-kabir.cos@knust.edu.gh

## Abstract

Glaucoma, characterized by chronic and progressive degeneration of retinal ganglion cells leading to a gradual decline in the visual field, poses a significant challenge in eye care delivery. Optometrists play a vital role in detecting and managing glaucoma. This study aims to determine how Ghanaian optometrists manage ocular adverse reactions (OADRs) in patients using antiglaucoma medications. A cross-sectional survey was conducted among 139 licensed optometrists using a pre-tested, semi-structured questionnaire. Data collected included socio-demographic characteristics, reported adverse reactions, and awareness of reporting or management protocols. Descriptive and inferential statistics were performed using SPSS version 25.0. Participants had a mean age of $31.6 \pm 5.5$ years and an average practice experience of $5.6 \pm 4.5$ years. Most respondents were female (71.2%), with 36.7% practicing in Greater Accra and 32.0% in private facilities. Nearly all participants (92.8%) reported encountering OADRs to AGMs. The most common symptoms were red eyes (36.0%), stinging (26.1%), burning sensation (25.4%), and itching (8.8%), while clinical signs included conjunctival hyperemia (45.5%), tear film deficiency (26.0%), and iris pigmentation changes (9.8%). Latanoprost (32.2%), timolol (23.7%), travoprost (17.6%), and brimonidine (10.2%) were the medications most frequently associated with these reactions. Optometrists primarily managed cases by switching to alternative medications (41.0%) or adjusting dosage and frequency (9.7%). However, 77.7% reported having no knowledge of existing protocols, and confidence in managing OADRs was moderately low (mean $= 2.67 \pm 0.49$ on a 5-point scale). No significant association was found between OADR reporting and either age (p = 0.616) or years of practice (p = 0.974). These findings highlight that although OADRs are common in glaucoma

**Data availability statement:** All relevant data are within the paper and its Supporting Information files.

**Funding:** The author(s) received no specific funding for this work.

**Competing interests:** The authors have declared that no competing interests exist.

management, optometrists in Ghana often lack standardized protocols and report low confidence in handling such cases. Targeted professional development and refresher training are needed to strengthen OADRs recognition, reporting, and management, ultimately improving patient eye care.

---

## Background

Glaucoma, characterized by the chronic and progressive degeneration of retinal ganglion cells leading to a gradual decline in visual field, poses a significant challenge in eyecare delivery [1,2]. The number of people diagnosed with glaucoma globally is projected to rise to 111.8 million in 2040, with Asia and Africa disproportionately affected [3]. The impact of glaucoma, particularly primary open-angle glaucoma, is notably pronounced among individuals of African descent [4]. In Ghana, glaucoma stands as a devastating condition, ranking second only to cataracts in its inexorable progression toward blindness [5]. The escalating prevalence of irreversible blindness in Ghana appears as a significant health concern.

Preserving the remaining visual field stands as the ultimate objective in managing glaucoma, with intraocular pressure (IOP) reduction identified as the sole modifiable factor for preventing associated visual field defects [6]. Various modalities, including topical therapy, oral medications, laser therapy, and surgery, are employed to lower IOP in glaucoma patients [7,8]. Among these treatment options, topical therapy emerge as the preferred choice due to their high efficacy [8,9]. However, these topical ophthalmic medications often comprise diverse therapeutic agents and additives, serving purposes such as preparation facilitation, solution/suspension stabilization, preservation, and product safety enhancement.

Despite the effectiveness of topical therapeutics in managing glaucoma, there is an escalating number of reports detailing adverse drug reactions among patients on antiglaucoma medication [10]. According to Kyei, Koffuor [11], there have been various verbal and informal reports of adverse drug reactions within the Ghanaian patient population and among ophthalmic eyecare practitioners. These associated adverse reactions have the potential to contribute to ocular discomfort, low compliance with antiglaucoma medications, and further deterioration in vision and the overall quality of life among glaucoma patients in Ghana.

Optometrists play a crucial role in the eyecare delivery, addressing vision impairment that affects an estimated 285 million patients worldwide [12]. In Ghana, the scope of practice for optometrists includes refraction and dispensing, diagnostics, and rehabilitation of visual system conditions [13]. Specifically, optometrists in Ghana employ a pharmacotherapeutic approach in the management of glaucoma and where necessary, make a referral to an ophthalmologist specialized in the treatment of glaucoma using laser or surgical approach. Their significant role highlights their substantial contribution to the treatment of glaucoma, prevention of blindness, and optimization of compromised eyesight, aligning with Sustainable Development Goal 3, which focuses on promoting health and well-being [14].

However, to date, it remains unclear the proportion of optometrists who receive and manage ocular adverse drug reactions associated with the use of antiglaucoma medications as well as what their preferred management pattern is for such reports. This novel study seeks to contribute to the understanding of how optometrists in Ghana manage ocular adverse reactions to antiglaucoma medications. It will provide valuable insights into the clinical practices of optometrists, which will be beneficial in developing the best practices and guidelines for the management of ocular adverse reactions to AGM and in enhancing patient care and safety.

## Methods

### Ethical statement

The research received approval from the Committee on Human Research, Publication, and Ethics at Kwame Nkrumah University of Science and Technology, Kumasi, Ghana (CHRPE/AP/1038/23), following formal institutional approval from the authority of Ghana Optometric Association. Written informed consent was obtained from all participants after explaining the objectives, nature, method, and importance of the study to them. The study adhered to the principles outlined in the Declaration of Helsinki. Our study obtained permission from the Ghana Optometric Association (GOA/RES/09/23). The study adhered to the ethical principles of anonymity, beneficence, non-maleficence, and justice. Data was coded without personal identifiers to ensure participants' anonymity, and all responses remained confidential. Participants faced no risks and had the right to withdraw or skip questions without consequences. These measures upheld ethical integrity and protected participants' rights and welfare.

### Study design, survey tool, setting and population

In this cross-sectional study conducted from February to May 2024 among practicing optometrists in Ghana, researchers employed a semi-structured questionnaire that was pre-tested on eight licensed optometrists and subsequently revised for clarity and ease of understanding. The questionnaire was administered both in person and online and covered key areas, including demographics, adverse ocular reactions to antiglaucoma medications, and knowledge of reporting systems for adverse medication reactions. Ghana is bordered by Côte d'Ivoire to the west, Burkina Faso to the north, Togo to the east, and the Gulf of Guinea to the south. The country has a population of 30.8 million people as of 2021, a growth rate of about 2.1%, and covers a total land area of roughly 238,533 km$^2$ [15]. Ghana is divided into 16 regions namely Greater Accra, Central, Western, Eastern, Ahafo, Ashanti, Volta, Northern, Upper East, Upper West, Bono East, Savannah, North – East, Oti, Western – North. The study population included all registered optometrists of the Ghana Optometric Association (GOA), who work across all regions. The GOA is the professional body that advances the field of optometry in Ghana and coordinates with the Allied Health Professions Council of Ghana (AHPC).

### Study participants

An official letter was sent to the Ghana Optometric Association (GOA) to explain the purpose of the study and request permission for the conduct of the study among its members. After review and further correspondence, the GOA leadership communicated the study details to all its members nationwide, encouraging participation. The GOA provided a professional registry of its members, which in 2024 listed 449 registered optometrists. Given our calculated sample size of 208, some participants were interviewed using hard-copy questionnaires during the association's annual general meeting. For those who could not be interviewed in person, online Microsoft forms of the questionnaire were provided using their correct contact information. One hundred and thirty-nine (139) optometrists completed the study questionnaires. Although the calculated sample size was 208, 139 optometrists eventually participated, due to limited availability and response rates.

### Data analysis

The survey data was processed and analyzed using Statistical Product and Service Solutions version 25.0, compatible with Windows 11. Data distribution was examined through descriptive statistics and cross-tabulations. Relationships between explanatory and response variables were evaluated using chi-square tests with significance set at $p \leq 0.05$.

## Results

### Demographic profile of the participants

A total of 139 optometrists (66.8% response rate) responded to the questionnaire. The mean age of the participants was 31.63 years (SD = ±5.54). On average, participants had 5.59 years (SD=±4.52) of practice experience. Among the participants, 71.2% were females, 36.7% practiced in the Greater Accra region, and 32.0% worked in private facilities. Table 1 presents a summary of the socio-demographic characteristics of the participants.

### Routine glaucoma management

A significant proportion of the optometrists (92.1%) reported managing glaucoma. Majority of the optometrists (92.8%) reported receiving complaints of ocular adverse drug reactions (OADR) to antiglaucoma medications (AGM). The incidence of OADRs to AGM was primarily identified through patient symptoms (54.2%) and clinical signs (44.4%) (see Table 2).

The mean age of those who managed glaucoma routinely was 31.70 years (SD = ±5.50), compared to 30.82 years (SD = ±6.27) for those who did not manage glaucoma routinely. The average years of practice among those who managed glaucoma routinely (5.59 years, SD = ± 4.50) was similar to those who did not (5.55 years, SD = ±4.97). There was no difference in age (p = 0.616) or years of practice (p = 0.974) between these two groups. Table 3 shows the crosstabulation of demographic characteristics of participants and routine glaucoma management and reports of Ocular ADRs.

### Symptoms, signs, and medications associated with ocular adverse drug reactions to anti-glaucoma medications

The most reported symptoms of OADR to AGM included red eyes (36.0%), stinging (26.1%) and burning sensation (25.4%) (Fig 1). The associated clinical signs found by the optometrists were conjunctival injection/hyperemia (45.5%) and tear film deficiency (26.0%) (Fig 2). The anti-glaucoma medications most often associated with these adverse reactions were latanoprost (32.2%), timolol (23.7%), travoprost (17.6%), and brimonidine (10.2%). Fig 3 shows the distribution of antiglaucoma medications associated with adverse drug reactions. Latanoprost was the drug most often associated with these symptoms, being implicated in over 60% of reports of red eyes, itchy eyes, and burning sensation (Table 4).

### Management practices, training, and reporting systems for ocular adverse drug reactions (OADR)

The primary management approaches employed by the optometrists included prescribing an alternative medication (41.0%) and adjusting the dosage or frequency (9.7%) of previously prescribed AGM. With 74.1% of the participants indicating no further training on the management of OADR to AGM, the confidence level of the participants in managing these ADRs was moderately low, with a mean score of 2.67 (+/-0.49) on a scale of 1 (lowest) and 5 (highest). Majority of the practitioners (77.7%) reported having no knowledge of existing system or protocol for reporting medications associated with patients' complaints of OADR. Table 5 shows management practices of adverse drug reactions with antiglaucoma medications, awareness and management training, and usage of reporting systems for OADRs.

## Discussion

The findings of our study highlight low reports of training and awareness among optometrists concerning the management and reporting of OADRs to AGM. A considerable proportion of optometrists (74.1%) indicated they had not received additional training on managing ocular adverse drug reactions associated with patient usage of antiglaucoma medications,

**Table 1. Demographic characteristics of participants.**

| Variables | Distribution |
|---|---|
| **a. Continuous variables** | **Mean (±SD)** |
| Age | 31.63 (±5.542) |
| Practice years | 5.59 (±4.523) |
| **b. Categorical variables** | **% (n)** |
| *Sex* | |
| Male | 28.8 (40) |
| Female | 71.2 (99) |
| *Practice location* | |
| Greater Accra | 36.7 (51) |
| Ashanti | 25.2 (35) |
| Central | 12.2 (17) |
| Eastern | 10.1 (14) |
| Western | 8.6 (12) |
| Western North | 0.0 (0) |
| Oti | 1.4 (2) |
| Volta | 0.0 (0) |
| Ahafo | 1.4 (2) |
| Bono | 1.4 (2) |
| Bono East | 0.7 (1) |
| Savannah | 0.0 (0) |
| Northern | 1.4 (2) |
| North East | 0.7 (1) |
| Upper East | 0.0 (0) |
| Upper West | 0.0 (0) |
| *Practice type* | |
| Private only | 32.0 (55) |
| CHAG_NGO | 12.8 (22) |
| Private with other optometrists | 29.7 (51) |
| Ghana Health Service | 10.5 (18) |
| Private with ophthalmologists and ophthalmic nurses | 6.4 (11) |
| Teaching Hospital | 3.5 (6) |
| Quasi-government, military, and university hospitals | 4.7 (8) |
| Consultancy | 0.6 (1) |

which correlates with their relatively low confidence levels, averaging a confidence score of 2.67. This lack of training is a critical issue, as it directly affects the ability of optometrists to effectively identify, assess, and manage OADRs. A substantial proportion of optometrists reported routinely managing glaucoma (91.2%) and receiving reports of OADRs (92.8%). The high percentage of optometrists involved in glaucoma management and the receipt of reports of OADRs underscores the importance of ensuring that these practitioners are well-trained in handling glaucoma and its associated adverse reactions. Since glaucoma management is a common practice among optometrists in Ghana [16], enhancing their knowledge and skills in this area could lead to better patient outcomes and more effective management of this chronic condition.

A significant proportion (92.8%) of optometrists indicated receiving reports of OADR to antiglaucoma medications. The most common symptoms included red eyes (36.0%), stinging (26.1%) and burning sensation (25.4%). The clinical signs observed were conjunctival injection/hyperemia (45.5%) and tear film deficiency (26.0%). These reports and signs

**Table 2. Adverse ocular reactions report rate and mode of diagnosing OADRs (n = 139).**

| Variables | Distribution |
|---|---|
| **a. Continuous variables** | **Mean (±SD)** |
| Frequency of ADR report | 2.61 (±0.830) |
| **b. Categorical variables** | **% (n)** |
| *Routine Glaucoma management* | |
| Yes | 92.1 (128) |
| No | 7.9 (11) |
| *Received reports of adverse drug reactions* | |
| Yes | 92.8 (129) |
| No | 7.2 (10) |
| *Diagnosing Ocular ADRs** | |
| Clinical signs | 44.4 (100) |
| Patient's symptoms | 54.2 (122) |
| Patient's history | 1.3 (3) |

Frequency of ADR reports was measured on a 5-point Likert scale, where higher scores indicate more frequent reporting of ADRs; *n ≠ 139, and the reported percentages may not total exactly 100% due to rounding.

**Table 3. Crosstabulation of demographic characteristics of participants and routine glaucoma management and reports of ocular ADRs.**

| Variables | Routine Glaucoma Management | | | Reports of Ocular ADRs | | |
|---|---|---|---|---|---|---|
| | **Yes** | **No** | | **Yes** | **No** | |
| **a. Continuous variables** | **Mean (±SD)** | **Mean (±SD)** | **p-value** | **Mean (±SD)** | **Mean (±SD)** | **p-value** |
| Age | 31.70 (±5.497) | 30.82 (±6.274) | 0.616 | 31.69 (±5.528) | 30.80 (±5.959) | 0.626 |
| Practice years | 5.59 (±4.504) | 5.55 (±4.967) | 0.974 | 5.59 (±4.554) | 5.60 (±4.326) | 0.993 |
| **b. Categorical variables** | **% (n)** | **% (n)** | | **% (n)** | **% (n)** | |
| *Sex* | | | | | | |
| Male | 71.1 (91) | 72.7 (8) | | 71.3 (92) | 70.0 (7) | |
| Female | 28.9 (37) | 27.3 (3) | | 28.7 (37) | 30.0 (3) | |
| *Practice location* | | | | | | |
| Greater Accra | 34.4 (44) | 63.6 (7) | | 36.4 (47) | 40.0 (4) | |
| Ashanti | 25.8 (33) | 18.2 (2) | | 23.3 (30) | 50.0 (5) | |
| Central | 12.5 (16) | 9.1 (1) | | 13.2 (17) | 0.0 (0) | |
| Eastern | 10.9 (14) | 0.0 (0) | | 10.9 (14) | 0.0 (0) | |
| Western | 8.6 (11) | 9.1 (1) | | 8.5 (11) | 10.0 (0) | |
| Western North | 0.0 (0) | 0.0 (0) | | 0.0 (0) | 0.0 (0) | |
| Oti | 1.6 (2) | 0.0 (0) | | 1.6 (2) | 0.0 (0) | |
| Volta | 0.0 (0) | 0.0 (0) | | 0.0 (0) | 0.0 (0) | |
| Ahafo | 1.6 (2) | 0.0 (0) | | 1.6 (2) | 0.0 (0) | |
| Bono | 1.6 (2) | 0.0 (0) | | 1.6 (2) | 0.0 (0) | |
| Bono East | 0.8 (1) | 0.0 (0) | | 0.8 (1) | 0.0 (0) | |
| Savannah | 0.0 (0) | 0.0 (0) | | 0.0 (0) | 0.0 (0) | |
| Northern | 1.6 (2) | 0.0 (0) | | 1.6 (2) | 0.0 (0) | |
| North East | 0.8 (1) | 0.0 (0) | | 0.8 (1) | 0.0 (0) | |
| Upper East | 0.0 (0) | 0.0 (0) | | 0.0 (0) | 0.0 (0) | |
| Upper West | 0.0 (0) | 0.0 (0) | | 0.0 (0) | 0.0 (0) | |

Age and years of practice are presented as continuous variables (mean ± SD) to show central tendency and variability. This allows for comparison across groups using independent sample t-tests. The reported percentages may not total exactly 100% due to rounding.

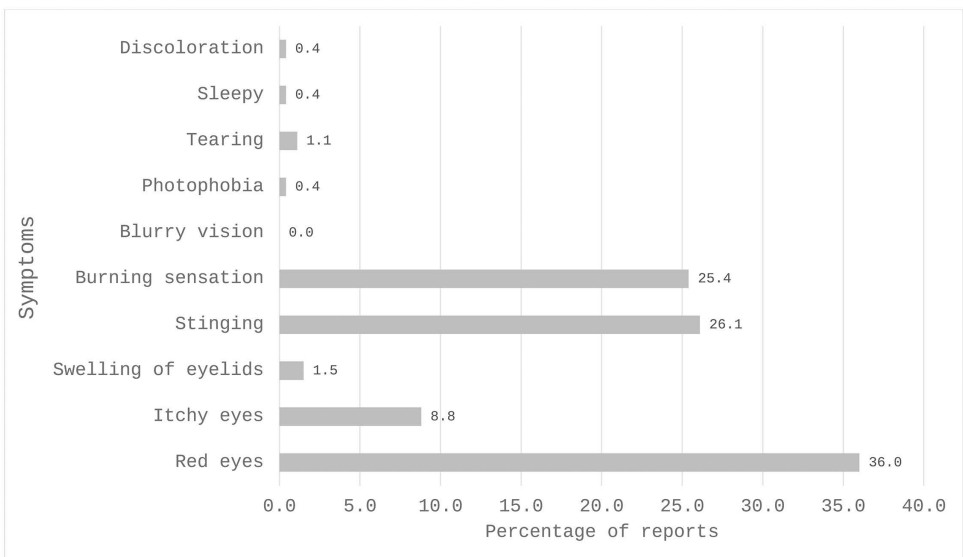

**Fig 1. Percentage distribution of patient reported symptoms of OADR to antiglaucoma medication.**

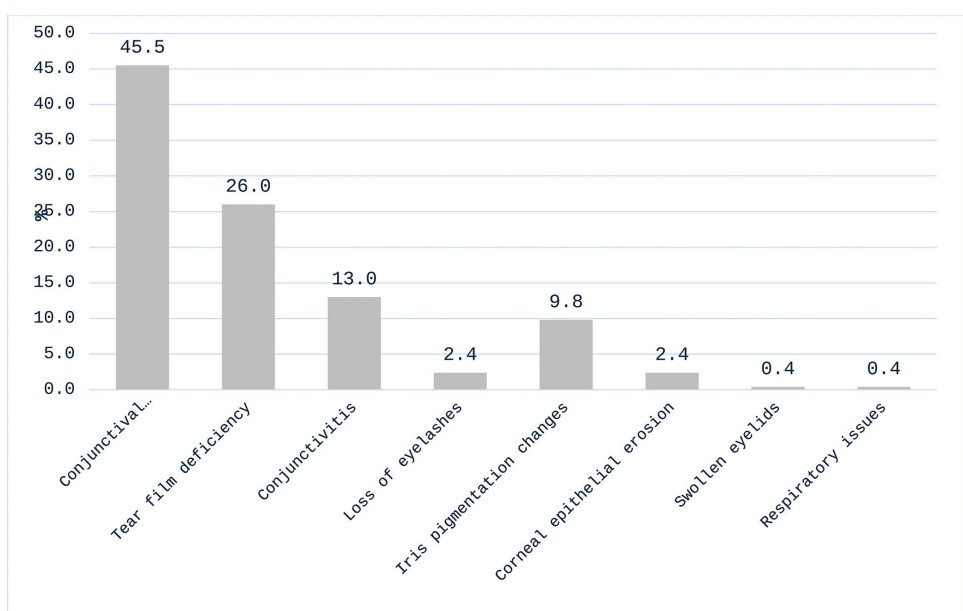

**Fig 2. Percentage distribution of commonest signs of ocular ADRs detected by optometrists.**

diagnosed were similar to the adverse reactions noted by Inoue [17] and Sidhan, Thankappan [18]. A Study by Wong, Wang [19] found adverse changes in tear film stability and conjunctival hyperemia in treated eyes, suggesting inflammatory mechanisms in dry eye development. Awe, Onakpoya [20] found a significantly higher prevalence of ocular surface disease among users of preserved topical anti-glaucoma medications, highlighting similar adverse reactions such as tear film instability and ocular surface staining in this present study. In another study by Sidhan, Thankappan [18], burning

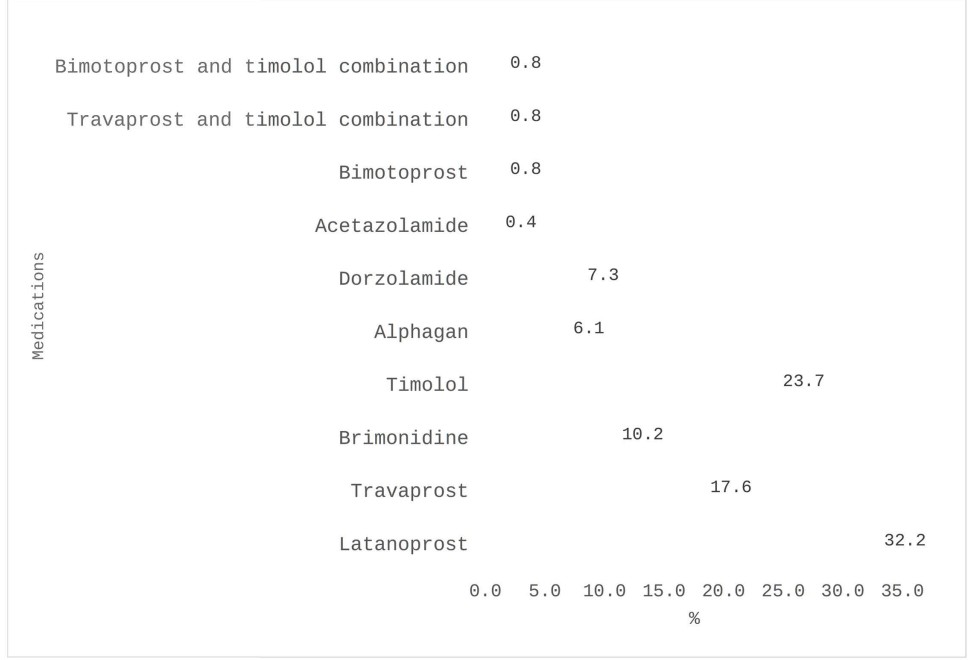

**Fig 3. Antiglaucoma medications associated with adverse drug reactions.**

**Table 4. Frequency distribution of adverse drug reactions (ADRs) associated with individual antiglaucoma medications.**

Antiglaucoma medications

| *ADR symptoms* | Latano-prost | Travaprost | Brimoni-dine | Timolol | Alphagan | Dorzol-amide | Acetazola-mide | Bimato-prost | Travaprost +timolol | Bimatop-rost +timolol |
|---|---|---|---|---|---|---|---|---|---|---|
| Red eyes | 63 (64.3%) | 35 (35.7%) | 20 (20.4%) | 41 (41.8%) | 10 (10.2%) | 14 (14.3%) | 0 (0.0%) | 2 (2.0%) | 2 (2.0%) | 1 (1.0%) |
| Itchy eyes | 16 (66.7%) | 7 (29.2%) | 7 (29.2%) | 16 (66.7%) | 3 (12.5%) | 4 (16.7%) | 0 (0.0%) | 0 (0.0%) | 0 (0.0%) | 0 (0.0%) |
| Swelling of eyelids | 2 (50.0%) | 2 (50.0%) | 2 (50.0%) | 3 (75.0%) | 0 (0.0%) | 1 (25.0%) | 0 (0.0%) | 0 (0.0%) | 0 (0.0%) | 0 (0.0%) |
| Stinging | 42 (59.2%) | 24 (33.8%) | 20 (28.2%) | 33 (46.5%) | 12 (16.9%) | 11 (15.5%) | 0 (0.0%) | 0 (0.0%) | 2 (2.8%) | 0 (0.0%) |
| Burning sensation | 44 (63.8%) | 24 (34.8%) | 15 (21.7%) | 35 (50.7%) | 8 (11.6%) | 10 (14.5%) | 0 (0.0%) | 1 (1.4%) | 1 (1.4%) | 1 (1.4%) |
| Blurry vision | 1 (100.0%) | 0 (0.0%) | 0 (0.0%) | 0 (0.0%) | 0 (0.0%) | 0 (0.0%) | 0 (0.0%) | 0 (0.0%) | 0 (0.0%) | 0 (0.0%) |
| Photophobia | 1 (100.0%) | 0 (0.0%) | 0 (0.0%) | 0 (0.0%) | 0 (0.0%) | 1 (100.0%) | 0 (0.0%) | 0 (0.0%) | 0 (0.0%) | 0 (0.0%) |
| Tearing | 1 (33.3%) | 2 (66.7%) | 0 (0.0%) | 1 (33.3%) | 0 (0.0%) | 0 (0.0%) | 0 (0.0%) | 0 (0.0%) | 0 (0.0%) | 0 (0.0%) |
| Sleepy | 0 (0.0%) | 0 (0.0%) | 0 (0.0%) | 1 (100.0%) | 0 (0.0%) | 0 (0.0%) | 0 (0.0%) | 0 (0.0%) | 0 (0.0%) | 0 (0.0%) |
| Discoloration | 1 (100.0%) | 1 (100.0%) | 0 (0.0%) | 0 (0.0%) | 0 (0.0%) | 0 (0.0%) | 0 (0.0%) | 0 (0.0%) | 0 (0.0%) | 0 (0.0%) |

Rows represent the presence of specific adverse drug reactions (ADRs), and columns represent the use (Yes") of the indicated medication. The cell values indicate the frequency (count) of ADR occurrence, with percentages calculated within each ADR category. The frequency of ADRs is presented separately for each medication to illustrate the specific adverse reactions associated with individual drugs.

sensation was reported as the most common adverse reaction (60.75%), aligning with our study findings. These symptoms experienced by patients on antiglaucoma medications may affect their adherence and compliance to the usage of antiglaucoma medications as reported in previous studies by Kaštelan, Tomić [21] and Abu Hussein, Eissa [22]. Given the common occurrence of symptoms and signs associated with AGM, optometrists must be equipped with the knowledge and tools to effectively address these issues.

PLOS Global Public Health

**Table 5. Management practices of adverse drug reactions with antiglaucoma medications, awareness and management training, and usage of reporting systems for OADRs.**

| Variables | Distribution |
|---|---|
| **a. Continuous variables** | **Mean (±SD)** |
| Confidence level of managing ocular ADRs | 2.67 (±0.485) |
| **b. Categorical variables** | **% (n)** |
| *Usual management approach for Ocular ADRs* | |
| Discontinuing medication | 28.8 (83) |
| Adjusting dosage/frequency | 9.7 (28) |
| Prescribing an alternative | 41.0 (118) |
| Referring to an ophthalmologist | 5.9 (17) |
| Monitor without changes | 8.7 (25) |
| Assure client to bear with it | 5.9 (17) |
| *Received training on Ocular ADR management* | |
| Yes | 25.9 (36) |
| No | 74.1 (103) |
| *Training on Ocular ADR management* | |
| Workshop | 27.5 (14) |
| Seminar | 35.3 (18) |
| Conference | 25.5 (13) |
| Online reading | 5.9 (3) |
| Academic lecture | 5.9 (3) |
| *Awareness of existing reporting systems for ocular ADRs* | |
| Yes | 31.2 (43) |
| No | 68.8 (95) |
| *Use of existing reporting systems for ocular ADRs* | |
| Yes | 22.3 (31) |
| No | 77.7 (108) |
| *Channel for submitting ADR reports* | |
| Food and Drugs Authority | 28.1 (9) |
| Pharmaceutical company | 59.4 (19) |
| Hospital pharmacy | 12.5 (4) |
| *Means of reporting ocular ADRs* | |
| Facility protocols/process | 80.5 (33) |
| Calling hotlines | 4.9 (2) |
| Mobile apps | 12.2 (5) |
| Company website | 2.4 (1) |

In our study, we found that most of the OADR were associated with latanoprost (32.2%) and timolol (23.7%). Previous studies have shown that prostaglandin analogues, particularly latanoprost, are the most frequently prescribed and persistent antiglaucoma medications [23,24]. It was reported in our study to be the major cause of ocular adverse reactions. This is in contrast to the study of Bhatt, Whittaker [25] which found that brimonidine was the most frequent agent causing adverse reactions, with 34.8% of patients experiencing the symptoms after more than 12 months of treatment. Although there were no reports of antiglaucoma medications causing lacrimal and Meibomian gland disorders in our study, Kam, Di Zazzo [26] highlighted that some of these antiglaucoma medications could cause dry eye disease through the dysfunction of the lacrimal and meibomian glands. A number of studies have pointed out that ocular surface disease is a common

comorbidity in glaucoma patients, exacerbated by preservatives like benzalkonium chloride (BAK) in topical medications since it is usually used in the preservation of most antiglaucoma medications [27–29]. Adverse reactions to preservatives can lead to discomfort, causing patients to drop their medications or use them inconsistently. A preservative-free formulations can significantly reduce the incidence of these adverse reactions, leading to better patient comfort and compliance with medication regimens as advocated by Baudouin [30], Bagnis, Papadia [31] and Lee, Wong [32].

In our study, optometrists primarily managed OADRs by prescribing alternative medications (41.0%) and adjusting dosage or frequency (9.7%) of previously prescribed AGM. Our findings are consistent with findings of McKee, Gupta [9] and Lusthaus and Goldberg [8], where most of the reported adverse reactions associated with topical medications were substituted with alternative eye drops. Thus, most of the optometrists preferred giving alternative eye drops and adjusting the dosage of the medications that the patients reported. This is consistent with the review guidelines set by Andole and Senthil [27] where the antiglaucoma medication causing the ocular adverse reaction is to be substituted with an oral or topical non-preserved and less toxic prepared form of the medication. Additionally, optometrists employed other strategies, including patient counseling and education on adherence to prescribed treatments, and monitoring the adverse effects of medications the patient complained of without changing the medication. By implementing these management strategies, optometrists aim to mitigate ocular adverse reactions associated with the use of antiglaucoma medications, improve patient compliance, and enhance therapeutic outcomes. This highlights the critical role of optometrists in ensuring the continuity and effectiveness of glaucoma care while minimizing patient discomfort and risk.

The findings of this study reveal significant gaps in further training among optometrists regarding the management of ocular adverse reactions to Antiglaucoma medications. A substantial majority of optometrists (74.1%) reported that they had not received any further training in managing OADRs, which is reflected in their moderately low confidence levels, with a mean confidence score of 2.67. This shows a lack of continuous professional development affecting their confidence in managing OADRs. In contrast, a study by Hahn, Friedman [33] on physicians shows that targeted training interventions can significantly improve practice, particularly in communication about medication adherence. This suggests that similar training for optometrists could enhance their confidence and competence in managing OADRs. Furthermore, our study shows that 77.7% of practitioners have no knowledge of existing systems for reporting OADRs. This is in line with the findings of Gordhon and Padayachee [34] who also reported significant underreporting of adverse drug reactions among surveyed patients in their studies. The underreporting in their study was attributed to a lack of knowledge and time constraints, similar to the issues identified in our study. The parallel between these findings suggests that the lack of further training and awareness of reporting systems for OADRs is a widespread problem that needs to be addressed comprehensively.

## Strengths of the study

One of the primary strengths of this study is its comprehensive assessment of the current practices and challenges faced by optometrists in managing and reporting ocular adverse drug reactions (OADR) in Ghana. The moderately high response rate ensures that the findings are representative of the population, providing robust data on the prevalence of OADR management and reporting practices. Additionally, the study's alignment with previous research, such as that by Gordhon and Padayachee [34] and Srisuriyachanchai, Cox [35], reinforces its relevance and underscores the importance of continuous professional development in pharmacovigilance. This consistency with prior findings lends credibility to the study's recommendations for improved training and education.

## Limitations of the study

This study has some limitations. The reliance on self-reported data may introduce bias, as participants might either overestimate their knowledge or underreport the ocular adverse reactions and challenges their clients face. Additionally, the cross-sectional nature of the study means it captures a single point in time, which may not fully represent trends or

changes over time. Finally, the study did not explore patient outcomes related to reports of OADRs, which could provide a more comprehensive understanding of the impact of these practices on patient care. Future research should consider longitudinal designs and direct patient outcome assessments to better understand the effectiveness of training and reporting systems.

## Implications and recommendations

This study highlights significant gaps in the awareness, training, and reporting of ocular adverse drug reactions among optometrists in Ghana, with important implications for practice, policy, and future research. The findings underscore the urgent need for evidence-based training programs that focus on the identification, management, and reporting of OADRs, particularly related to antiglaucoma medications. Such training will address the prevalent lack of confidence and competence among optometrists, ensuring timely and effective patient care. The study also points to the absence of standardized reporting systems for OADRs, urging policymakers to prioritize the development of structured protocols to improve pharmacovigilance in optometric practice. Further research is needed to evaluate the effectiveness of these training programs and the impact of standardized reporting systems on the accuracy and timeliness of OADR documentation. Additionally, the Ghana Optometric Association, regulatory bodies, and academic institutions should collaborate to integrate OADR management into professional development curricula and organize regular workshops to enhance ongoing education and awareness. By addressing these issues, this study provides a framework for improving optometric practice, enhancing patient safety, and contributing to the development of a more robust pharmacovigilance system in Ghana.

## Conclusion

The findings of our study reveal substantial gaps in knowledge and confidence among optometrists, emphasizing the necessity of continuous professional education in pharmacovigilance. Addressing these educational gaps is crucial for mitigating the adverse effects of anti-glaucoma medications and improving patient outcomes. The study underscores the importance of refresher courses and ongoing professional development in managing ocular adverse reactions to anti-glaucoma medications.

## Supporting information

**S1 Data. Dataset used for analysis.**
(SAV)

## Acknowledgments

We would like to appreciate Dr. Richchris Konadu Yiadom, Dr. Francisca Odoi, Dr. Sylvester Kyeremeh, and the entire body of Ghana Optometric Association for their immense support. We also extend our appreciation to all the optometrists in Ghana who took part in this study.

Patient and public involvement: Patients and/or the public were not involved in the design, conduct, reporting, or dissemination plans of this research.

Participant consent: All participants agreed and gave their consent to participate in the study after the aims and potential benefits and risks were well spelt out.

## Author contributions

**Conceptualization:** Abdul-Kabir Mohammed, Prince Mintah.

**Data curation:** Stephanie Obeng-Inkoom.

**Formal analysis:** Prince Mintah, Michael Kwesi Asante, Gabriel Kwaku Agbeshie.

**Methodology:** Abdul-Kabir Mohammed, Michael Kwesi Asante.

**Project administration:** Agnes Oppong, Stephanie Obeng-Inkoom.

**Resources:** James Forson, Agnes Oppong.

**Software:** Prince Mintah, Gabriel Kwaku Agbeshie.

**Supervision:** Agnes Oppong.

**Visualization:** James Forson.

**Writing – original draft:** Michael Kwesi Asante, James Anin Odame.

**Writing – review & editing:** Abdul-Kabir Mohammed, James Anin Odame.

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
