## [Decision Letter · Decision Letter 0]

29 Dec 2024

PGPH-D-24-02595

MANAGEMENT OF OCULAR ADVERSE REACTIONS TO ANTIGLAUCOMA MEDICATIONS – SURVEY OF OPTOMETRISTS IN GHANA

Dear Dr. Mohammed,

Thank you for submitting your manuscript to PLOS Global Public Health. After careful consideration, we feel that it has merit but does not fully meet PLOS Global Public Health’s publication criteria as it currently stands. Therefore, we invite you to submit a revised version of the manuscript that addresses the points raised during the review process.

We look forward to receiving your revised manuscript.

Kind regards,

Akhilanand Chaurasia

Academic Editor

Journal Requirements:

1. Please provide additional details regarding participant consent. In the ethics statement in the Methods and online submission information, please ensure that you have specified (1) whether consent was informed and (2) what type you obtained (for instance, written or verbal, and if verbal, how it was documented and witnessed).

3. Please provide separate figure files in .tif or .eps format.

4. Please provide an Author Summary. This should appear in your manuscript between the Abstract (if applicable) and the Introduction, and should be 150–200 words long. The aim should be to make your findings accessible to a wide audience that includes both scientists and non-scientists. Sample summaries can be found on our website under Submission Guidelines: 

https://journals.plos.org/globalpublichealth/s/submission-guidelines#loc-parts-of-a-submission

Additional Editor Comments (if provided):

Dear Author,

Rebuttal of queries raised by reviewers are highly recommended.

Reviewers' comments:

Reviewer's Responses to Questions

**Comments to the Author**

1. Does this manuscript meet PLOS Global Public Health’s publication criteria?

Reviewer #1: Yes

Reviewer #2: No

2. Has the statistical analysis been performed appropriately and rigorously?

Reviewer #1: Yes

Reviewer #2: No

3. Have the authors made all data underlying the findings in their manuscript fully available (please refer to the Data Availability Statement at the start of the manuscript PDF file)?

Reviewer #1: Yes

Reviewer #2: Yes

4. Is the manuscript presented in an intelligible fashion and written in standard English?

Reviewer #1: Yes

Reviewer #2: Yes

Reviewer #1: 1. In line 75, it should be ‘’there’’ instead of ‘’here’’ after According to Kyei, Koffuor (11)

2. SPSS has changed its name to Statistical Product and Service Solutions instead of Statistical Package for the Social Sciences

3. In line 121 and 122, evaluating relationships between explanatory and response 122 variables and maintaining significance at p ≤ 0.05 is an incomplete statement . What statistical method of analysis was used in the evaluation

4. Explain how the ethical principles of anonymity, beneficence, malficence and Justice were applied in this study.

5. In line 327, it should be ---our findings are consistent and NOT our findings is consistent

6. What makes lack of further training and awareness of reporting systems for OADRs is a widespread problem. Substantiate this assertion with evidences

7. The research implications in this study are week. Research implications are the consequences of research findings. They go beyond results. Research implication should inform further research, shape policy, or spark new solutions to old problems. Research Implications should link to specific results within your research to ensure they’re grounded in reality. Research implications need to be rational and based on data from your research, not conjecture. An evidence-based approach to implications will lend credibility and validity to your work.

Reviewer #2: 1. As per the title of the manuscript "MANAGEMENT OF OCULAR ADVERSE REACTIONS TO ANTIGLAUCOMA

MEDICATIONS – SURVEY OF OPTOMETRISTS IN GHANA" here the management is missing in the manuscript so it is not at all justifying the title.

2. The adverse reaction symptoms data are collected from the optometrist not from the patient directly it may create bias.

3. semi-structured questionnaire are not attached and any validation by authorities person are missing.

4. p- value is not significant.

**Do you want your identity to be public for this peer review?** For information about this choice, including consent withdrawal, please see our Privacy Policy

Reviewer #1: **Yes: ** IBRAHEEM SHOLA ABDULRAHEEM

Reviewer #2: No

---

## [Decision Letter · Decision Letter 1]

8 May 2025

PGPH-D-24-02595R1

MANAGEMENT OF OCULAR ADVERSE REACTIONS TO ANTIGLAUCOMA MEDICATIONS – SURVEY OF OPTOMETRISTS IN GHANA

Dear Dr. Mohammed,

Thank you for submitting your manuscript to PLOS Global Public Health. After careful consideration, we feel that it has merit but does not fully meet PLOS Global Public Health’s publication criteria as it currently stands. Therefore, we invite you to submit a revised version of the manuscript that addresses the points raised during the review process.

Two reviewers have assessed your manuscript and provided some brief comments below. Please carefully revise your manuscript to address the suggestions made by Reviewer 3.

We look forward to receiving your revised manuscript.

Kind regards,

Sarah Jose, Ph.D.

Staff Editor

Journal Requirements:

1. Please provide additional details regarding participant consent. In the ethics statement in the Methods and online submission information, please ensure that you have specified (1) whether consent was informed and (2) what type you obtained (for instance, written or verbal, and if verbal, how it was documented and witnessed).

Additional Editor Comments (if provided):

Reviewers' comments:

Reviewer's Responses to Questions

**Comments to the Author**

Reviewer #1: All comments have been addressed

Reviewer #3: All comments have been addressed

publication criteria?

Reviewer #1: Yes

Reviewer #3: Yes

3. Has the statistical analysis been performed appropriately and rigorously?

Reviewer #1: Yes

Reviewer #3: Yes

4. Have the authors made all data underlying the findings in their manuscript fully available (please refer to the Data Availability Statement at the start of the manuscript PDF file)?

Reviewer #1: Yes

Reviewer #3: Yes

5. Is the manuscript presented in an intelligible fashion and written in standard English?

Reviewer #1: Yes

Reviewer #3: Yes

Reviewer #1: None. This paper has been read before and all corrections have been effected

Reviewer #3: Add a brief description of how the questionnaire was pre-tested (e.g., pilot testing with a sample group, expert review by ophthalmologists/statisticians) in the Methods section

**Do you want your identity to be public for this peer review?** For information about this choice, including consent withdrawal, please see our Privacy Policy

Reviewer #1: **Yes: ** IBRAHEEM SHOLA ABDULRAHEEM

Reviewer #3: No

---

## [Decision Letter · Decision Letter 2]

24 Aug 2025

PGPH-D-24-02595R2

MANAGEMENT OF OCULAR ADVERSE REACTIONS TO ANTIGLAUCOMA MEDICATIONS – SURVEY OF OPTOMETRISTS IN GHANA

Dear Dr. Mohammed,

Thank you for submitting your manuscript to PLOS Global Public Health. After careful consideration, we feel that it has merit but does not fully meet PLOS Global Public Health’s publication criteria as it currently stands. Therefore, we invite you to submit a revised version of the manuscript that addresses the points raised during the review process.

We look forward to receiving your revised manuscript.

Kind regards,

Helen Howard

Staff Editor

Journal Requirements:

Reviewers' comments:

Reviewer's Responses to Questions

**Comments to the Author**

Reviewer #3: All comments have been addressed

Reviewer #4: All comments have been addressed

publication criteria?

Reviewer #3: Yes

Reviewer #4: Yes

3. Has the statistical analysis been performed appropriately and rigorously?

Reviewer #3: Yes

Reviewer #4: Yes

4. Have the authors made all data underlying the findings in their manuscript fully available (please refer to the Data Availability Statement at the start of the manuscript PDF file)?

Reviewer #3: Yes

Reviewer #4: Yes

5. Is the manuscript presented in an intelligible fashion and written in standard English?

Reviewer #3: Yes

Reviewer #4: No

Reviewer #3: I do not have additional comments

Reviewer #4: In methodology, pretested questionnaire is used but validation result is not stated.

Study duration not mentioned in methodology part.

Clarity on sample size is required as the calculated sample size given is 208 but collected participants number is 139 which needs explanation.

In result section, Table 3. The percentage given for categorical variables need to be checked carefully. Little more clarity on explanation is needed for both Table 2 & 3. Continuous variable on Table 2 needs explanation.

In figure 1. reported symptom of OADR to AGM was shown to as red eyes be the most followed by stinging and burning sensation, but in results section no red eyes are mention, other than red eyes, it was written as stinging and burning sensation to be the most.

It will be good if given the details of ADR produced by individual drug with frequency.

In statistical analysis part, only one test has been performed Outcome analysis if performed will impact the study.

For overall language, Minor English editing is required.

**Do you want your identity to be public for this peer review?** For information about this choice, including consent withdrawal, please see our Privacy Policy

Reviewer #3: No

Reviewer #4: **Yes: ** Dr. Sreeja PA

---

## [Decision Letter · Decision Letter 3]

16 Sep 2025

MANAGEMENT OF OCULAR ADVERSE REACTIONS TO ANTIGLAUCOMA MEDICATIONS – SURVEY OF OPTOMETRISTS IN GHANA

PGPH-D-24-02595R3

Dear Prof Mohammed,

We are pleased to inform you that your manuscript 'MANAGEMENT OF OCULAR ADVERSE REACTIONS TO ANTIGLAUCOMA MEDICATIONS – SURVEY OF OPTOMETRISTS IN GHANA' has been provisionally accepted for publication in PLOS Global Public Health.

Best regards,

Julia Robinson

Executive Editor

Reviewer #4:

Reviewer Comments (if any, and for reference):

Reviewer's Responses to Questions

**Comments to the Author**

Reviewer #4: All comments have been addressed

publication criteria?

Reviewer #4: Yes

3. Has the statistical analysis been performed appropriately and rigorously?

Reviewer #4: Yes

4. Have the authors made all data underlying the findings in their manuscript fully available (please refer to the Data Availability Statement at the start of the manuscript PDF file)?

Reviewer #4: Yes

5. Is the manuscript presented in an intelligible fashion and written in standard English?

Reviewer #4: Yes

Reviewer #4: There was an issue with the sample size which has been addressed by author

**Do you want your identity to be public for this peer review?** For information about this choice, including consent withdrawal, please see our Privacy Policy

Reviewer #4: No
